# Use of Flowering Plants to Enhance Parasitism and Predation Rates on Two Squash Bug Species *Anasa tristis* and *Anasa armigera* (Hemiptera: Coreidae)

**DOI:** 10.3390/insects10100318

**Published:** 2019-09-25

**Authors:** Mary L. Cornelius, Bryan T. Vinyard, Michael W. Gates

**Affiliations:** 1Invasive Insect Biocontrol and Behavior Lab, ARS-USDA, Beltsville Agriculture Research Center, 10300 Baltimore Ave., Bldg. 007, Beltsville, MD 20705, USA; 2Statistics Group, USDA Agricultural Research Service, Northeast Area Office, Beltsville, MD 20705, USA; bryan.vinyard@ars.usda.gov; 3Systematic Entomology Laboratory, ARS-USDA, c/o National Museum of Natural History, Smithsonian Institution, 10th & Constitution Ave. NW, MRC 168, Washington, DC 20560, USA; michael.gates@ars.usda.gov

**Keywords:** *Gryon pennsylvanicum*, *Ooencyrtus anasae*, *Trichopoda pennipes*, biological control, floral resources, cucurbit pests, natural enemies

## Abstract

A two-year study evaluated the effect of a flowering border of buckwheat *Fagopyrum esculentum* Moench on rates of egg parasitism, egg predation and adult parasitism on two squash bug species, *Anasa tristis* (DeGeer) and *Anasa armigera* Say, by comparing rates in squash fields with and without a flowering border. Furthermore, we evaluated whether there was an edge effect by comparing parasitism and predation rates in plots located in the corner of a squash field with plots located in the center of a squash field for fields with and without a flowering border. The egg parasitism rates were not affected by either treatment (flowering border or control) or plot location (edge or center). *Anasa armigera* egg masses only accounted for 4.3% of the total egg masses collected. The egg parasitism rates increased gradually throughout the season, peaking in the last week of August in 2017 at 45% for *A. tristis* egg masses. The most common egg parasitoid recovered was *Gryon pennsylvanicum* (Ashmead) followed by *Ooencyrtus anasae* (Ashmead). Adult parasitism was not affected by treatment, but *A. tristis* adult parasitism rates were higher in plots located on the edge of squash fields compared with plots located in the center of squash fields in 2016. Since adult parasitoid, *Trichopoda pennipes* (Fabricius) flies were observed visiting buckwheat flowers, future studies could explore the possibility that the flowering buckwheat may have a more impact on adult parasitism if there was a greater distance between fields with and without a flowering border.

## 1. Introduction

The loss of habitat and conversion of land to urbanization and intensive agriculture are major factors causing the loss of insect biodiversity [1]. Urban farms and gardens can increase biodiversity, ecosystem services, and climate resilience [2]. As urbanization increases, there is an increasing interest in the use of land for urban and peri-urban agriculture as a means of maintaining biodiversity and providing products to local communities [2]. The use of floral resources in urban farms can increase ecosystem services such as pollination and biological control [3,4]. Although floral resources have the potential to benefit farmers in both urban and rural areas, urban farmers are often growing organic produce for sale to local communities through farmers’ markets [2]. Therefore, urban farmers have a vested interest in finding methods of biological control to manage pest populations without a reliance on pesticides.

The use of floral resources in cropping systems has been shown to increase parasitoid longevity, fecundity, and retention [5,6,7,8,9,10]. For example, a two-year field study compared parasitism rates by *Trichopoda pennipes* (Diptera: Tachinidae) on hemipteran adults in peanut-cotton plots with and without flowering milkweed *Asclepias curassavica* L. (Gentianales: Apocynaceae). In the first year of the study, parasitism of *Nezara viridula* (L.) (Hemiptera: Pentatomidae) was significantly higher in plots with milkweed (61.6%) than in control plots (13.3%). In the second year, parasitism of *Leptoglossus phyllopus* (L.) (Hemiptera: Coreidae) by *T. pennipes* was significantly higher in plots with milkweed (24.0%) than in control plots (1.1%) [11].

Floral resources can also attract generalist predators [12]. Flowering alyssum increased the abundance of generalist predators of aphids in apple orchards [13]. The abundance of hoverflies, lacewings, and ladybird beetles was increased by using strips with a mixture of flowering plants [14]. Cornflowers *Centaurea cyanus* L. (Asterales: Asteraceae) increased predation on eggs of the cabbage moth *Mamestra brassicae* (Linnaeus) [9,15]. 

Several studies have examined the use of floral resources to control pests in cucurbit crops. The addition of a strip of either flowering alyssum or a mixture of perennial flowering plants in pumpkin fields increased egg predation on spotted cucumber beetles *Diabrotica undecimpunctata* (Linnaeus) (Coleoptera: Chrysomelidae), but not squash bugs *Anasa tristis* (De Geer) (Hemiptera: Coreidae) [16]. Fair and Braman [17] evaluated the effects of floral resources on squash bug populations with mixed results. The addition of floral resources decreased squash bug populations in four of eight possible year, site, and planting date combinations, but squash bug populations were higher in plots with floral resources in some cases. The use of companion planting with flowering plants did not significantly reduce squash bug populations [18]. Flowering strips increased the abundance of beneficial insects but did not increase cucumber yields [19].

The nectar of flowering buckwheat *Fagopyrum esculentum* Moench (Polygonaceae: Polygonales) is attractive to parasitoids and can increase their longevity and fecundity [20,21,22,23,24]. For example, when the aphid parasitoid *Diaeretiella rapae* (M’Intosh) (Hymenoptera: Braconidae) was exposed to flowering buckwheat in the laboratory, it survived 4–5 times longer than the water-only control and 2–3 times longer than parasitoids exposed to flowering alyssum [10]. However, field studies on the use of buckwheat to enhance parasitism rates have had mixed results. Flowering buckwheat in soybean enhanced parasitism of *Euschistus servus* (Say) (Hemiptera: Pentatomidae) egg masses by *Telenomus podisi* Ashmead (Hymenoptera: Scelionidae) in cotton [25]. In a study using buckwheat as a border around cabbage plots, Lee and Heimpel [26] found that parasitism on *Trichoplusia ni* (Hübner) (Noctuidae: Lepidoptera) by the larval parasitoid, *Voria ruralis* (Falléen) (Diptera: Tachinidae) and parasitism on *Pieris rapae* (L.) (Lepidoptera: Pieridae) by the larval parasitoid *Cotesia rubecula* (Marshall) (Hymenoptera: Braconidae) were higher in plots with a buckwheat border than in control plots. However, parasitism by *Euplectrus plathypenae* (Howard) (Hymenoptera: Eulophidae) on *T. ni* was higher in control plots than in plots with a buckwheat border. At one vineyard, the use of buckwheat increased parasitism on a complex of leafrollers (Lepidoptera: Tortricidae) by 50%, whereas buckwheat had no effect on parasitism rates at the other vineyard studied [27]. 

Flowering buckwheat has been used as a living mulch in squash fields to reduce pest populations, increase natural enemy populations, and improve yields [28,29,30]. However, none of these studies examined the effects of buckwheat on biological control agents of squash bugs. In this study, we evaluated whether the use of flowering buckwheat as a border around squash fields increased the rates of parasitism and predation on egg masses and adults of squash bugs, *Anasa tristis* (DeGeer) and *Anasa armigera* Say (Hemiptera: Coreidae). *Anasa tristis* is a serious pest of cucurbit crops and acts as a vector of cucurbit yellow vine disease which can severely damage crops [31,32,33,34]. It is found throughout North America [34]. Squash bugs overwinter as adults and emerge in the spring. Squash bugs complete their entire lifecycle in 6–8 weeks and can complete one to three generations per year, depending on location [34]. There has been limited research on the biology or natural enemies of *A. armigera*. It is a minor pest of cucurbit crops [35,36,37]. It is present in squash fields at the Beltsville Agricultural Research Center along with *A. tristis* [38]. In a recent laboratory study on the oviposition behavior of squash bugs, *A. armigera* was reared successfully on both yellow squash, *Cucurbita pepo* L. (Cucurbitales: Cucurbitaceae) and cucumber, *Cucumis sativus* L. (Cucurbitales: Cucurbitaceae). In paired choice tests, *A. armigera* was equally likely to oviposit on *C. pepo* or *C. sativus*, regardless of the species it was reared on [39].

There are two commonly occurring parasitoids of the common squash bug, *A. tristis*, the solitary egg parasitoid, *Gryon pennsylvanicum* (Ashmead) (Hymenoptera: Scelionidae) [40,41] and the adult parasitoid, *Trichopoda pennipes* (Fabricius) (Diptera: Tachinidae) [42,43]. In a two-year study in Maryland, *G. pennsylvanicum* accounted for over 99% of egg parasitism and the average rate of parasitoid emergence peaked on wild egg masses on the fifth week of July at 72.8% [44]. Reported rates of adult parasitism by *T. pennipes* were 12% in Virginia [34], 20% in Connecticut [43], and 20–30% in Kentucky [45]. Although there are no published field studies of parasitism on *A. armigera*, there was no difference in host acceptance by *G. pennsylvanicum* of egg masses of *A. tristis* and *A. armigera* in laboratory choice tests [38]. Predators of squash bug eggs include Araneae, Formicidae and Gryllidae [16]. Molecular analysis determined that Geocoridae, Coccinellidae, Nabidae, and web building and hunting spiders consumed squash bugs. However, it is not known which life stages they were preying upon [46]. In field tests in Kentucky, egg predation of *A. tristis* was low, ranging from 2.2% to 7.2% [45]. In Maryland, average predation of naturally occurring *A. tristis* eggs was 5.2% in a two-year study [44].

This study evaluated the use of floral resources in squash fields planted at an agricultural research center located in an urban area. This study compared rates of egg parasitism, egg predation, and adult parasitism of *A. tristis* and *A. armigera* in squash fields with a flowering buckwheat border to rates in control squash fields without a flowering border. We also evaluated whether the rates of egg parasitism, egg predation and adult parasitism varied between plots located at the edge of the field and plots located in the center of the field to determine if natural enemies attracted to flowering buckwheat were more abundant near the borders than in the center of the squash field.

## 2. Materials and Methods

### 2.1. Field Experiment

In 2016 and 2017, field tests were conducted in an urban location at the Beltsville Agricultural Research Center (BARC), Beltsville, MD. Squash fields were planted using *Cucurbita pepo* cv. Slick Pik YS26′ where buckwheat, *F. esculentum*, was planted at a rate of approximately 45 kg/ha in a three-meter-wide border around the treated squash fields compared with control squash fields where a three-meter-wide border was mowed (Figure 1). Each year, four squash fields were planted from seed at a rate of 3.36 kg/ha in 0.76 m rows 12–14 June 2016 and 28–30 June 2017. Planting was delayed in 2017 due to heavy rains. There were two blocks, at least 700 m apart, with a treated and a control field in each block. Squash fields were planted in different locations within the same two blocks each year. In both years, the treated and control fields were at least 50 m apart. Each field had two plots, a center plot and an edge plot. Each plot was 10 × 10 m with at least a 15-m distance between plots. There were approximately 400 squash plants in each plot. The center plot was located in the center of the field and the edge plot was located in a corner of the field so that two sides of the plot were adjacent to the border of the field. Squash fields were located adjacent to soybean, corn, and pepper fields.

Once the buckwheat plants started blooming, every squash plant in each experimental plot was checked once a week for the presence of naturally occurring squash bug egg masses and adult squash bugs of both species. In 2016, plots were checked weekly from 27 June–19 August. In 2017, plots were checked weekly from 17 July–26 August. The buckwheat plants continued to bloom throughout the experiment. All egg masses and adults found in the experimental plots were collected and brought into the laboratory. The number of eggs in each egg mass was counted. Egg masses were checked for signs of predation that leave visible evidence of damage on eggs. Egg masses were kept in Petri dishes in an incubator (25 °C; 16:8 [L:D]) until either nymphs or parasitoids emerged. Eggs with no emergence were dissected and unemerged nymphs or adult parasitoids were counted. Parasitism rates were calculated by combining the number of emerged and unemerged parasitoids. The rates of nymph eclosion, parasitism, predation, or unknown egg mortality were calculated by dividing the number of eggs in each category by the total number of eggs in each egg mass.

Adult squash bugs were maintained in an incubator (25 °C; 16:8 [L:D]) for up to a month in cylindrical plastic containers (19.3 cm height by 20.0 cm diameter) (Pioneer Plastics, Inc., Eagan, MN, USA). Each container had a hole (approximately 14 cm in diameter) in the lid covered with a fine mesh screen, a filter paper on the bottom, and two cotton plugged glass shell vials filled with water attached with a rubber band to prevent them from rolling. Adults were maintained on fresh cuttings of cucurbit plants grown in a greenhouse. After the death of an adult squash bug, the container was checked for the presence of a tachinid fly pupa. The pupa was placed in a Petri dish and kept in the incubator until the emergence of the adult tachinid fly. Tachinid parasitoids were kept in a vial of alcohol as voucher specimens.

Ethanol-preserved specimens were dehydrated through increasing concentrations of ethanol and transferred to hexamethyldisilazane (HMDS) before point-mounting. MWG identified *Ooencyrtus anasae* using a Leica M205C stereomicroscope with 10 × oculars and a Leica LED ring light source was used for point-mounted specimen observation [47]. All specimens were determined to genus by sight identification or using Gibson et al. [48]. Specimens were identified to species using relevant keys and primary literature [49]. All species identifications were corroborated through comparison with authoritatively identified specimens in the Smithsonian National Museum of Natural History.

### 2.2. Statistical Analysis

Comparisons of egg masses of the two squash bug species were conducted based only on data when both species were observed in 2017 from July week 3–August week 4. Treatment (flowering border or control), plot location (edge plots or center plots) and year (2016 or 2017) effects were examined based only on *A. tristis* data observed in both years from July week 3–August week 2. Nymph eclosion, eggs parasitized, and egg mortality rates were each modeled using a generalized linear mixed effects negative binomial ANOVA with offset log eggs per mass. Species comparisons were conducted within each treatment by plot location by week. Treatment by plot location comparisons was conducted within each year by week. Adult parasitism rates were evaluated for treatment, plot location, and treatment by plot location interaction effects within each squash bug species using a generalized linear mixed effects negative binomial ANOVA with offset log squash bug adults. Statistical analyses were accomplished using SAS v9.4 PROC GLIMMIX (SAS Institute Inc.: Cary, NC, USA) [50] and pdmix800.sas (SAS Institute: Cary, NC, USA) macro to obtain means comparisons letter groups [51].

## 3. Results

### 3.1. Field Collections of Egg Masses and Adults

A total of 1176 egg masses was collected from the last week of June until the last week of August in 2016 and 2017, 95.7% *A. tristis* egg masses and 4.3% *A. armigera* egg masses. The mean number (±SE) of *A. tristis* eggs per mass was 16.9 ± 2.2 with a range of 1–44 eggs per mass. The mean number (±SE) of *A. armigera* eggs per mass was 11.0 ± 9.3 with a range of 2–37 eggs per mass. Squash bug adults started laying eggs in squash fields within two weeks of planting. In 2016, weekly collections of *A. tristis* egg masses peaked with 138 egg masses collected in the third week of July (Figure 2A). In 2017, weekly collections of *A. tristis* egg masses peaked in the second week of August at 133 egg masses (Figure 2B). Although collections of *A. armigera* egg masses were very low in both 2016 and 2017, the number collected in 2017 was higher and peaked in the final week of August at 15 egg masses (Figure 2). 

*Anasa armigera* comprised 56.1% of the 130 adults collected in 2016 and 2017. The number of *A. tristis* adults collected was higher than the number of *A. armigera* adults collected in 2016 (Figure 3A), but the number of *A. armigera* adults collected was higher than *A. tristis* adults in 2017 (Figure 3B). Most *A. armigera* adults were collected in August of 2017. No adults were collected in the final week in 2017 (Figure 3B).

### 3.2. Egg Parasitism, Egg Predation, and Nymph Eclosion

Of the 2158 eggs that were parasitized, 93.9% were parasitized by *G. pennsylvanicum* and 5.9% were parasitized by the gregarious egg parasitoid *Ooencyrtus anasae* (Ashmead) (Hymenoptera: Encyrtidae) and 0.09% were parasitized by *Anastatus* sp. (Hymenoptera: Eupelmidae). 

Egg predation was too low to evaluate statistically. A total of 28 eggs (0.14%) were damaged by predators out of 19,640 total eggs collected. However, 21 of those 28 damaged eggs were collected in squash fields with a flowering border.

In a comparison of egg masses of the two squash bug species, there were no significant differences detected in the nymph eclosion rate, egg parasitism rate, or the rate of egg mortality from unknown causes between fields with and without a flowering border. The overall egg parasitism rate was a mean ± SE of 11.9 ± 0.8% for *A. tristis* eggs and 12.6 ± 3.7 for *A. armigera* egg masses. 

Since the overall number of *A. armigera* egg masses collected was very low (4.3%), the analysis of treatment, plot location, week and year were performed separately for *A. tristis* egg masses. There were no significant differences in *A. tristis* egg parasitism for treatment, plot location, or year (Figure 4).

However, there were significant weekly differences in egg parasitism rates and there were significant interactions effects for egg parasitism (Table 1). There were also significant four-way interaction effects for treatment by plot location by year and by week for egg parasitism. In 2016 and 2017, egg parasitism rates were significantly different between edge plots in control and treatment fields in the third week of July (Figure 5a,b). In 2017, egg parasitism rates were significantly different between center and edge plots in control fields in the first week of August (Figure 5B). In 2017, there were two incidences where only a single *A. tristis* egg mass was collected in a specific plot location during weekly collections. In July week 4, a single egg mass was collected in the edge plots in control fields and *G. pennsylvanicum* emerged from 100% of those eggs. In August week 1, a single egg mass was collected from center plots in control fields and parasitoids emerged from 18.2% of those eggs (Figure 5B). 

Nymph eclosion rates were significantly higher in squash fields with a flowering border with a mean ± SE of 79.8 ± 1.2% compared with 75.2 ± 1.6% in control fields and significantly higher in center plots (79.6 ± 1.5%) than in edge plots (77.2 ± 1.3%). There were significant weekly differences in nymph eclosion rates (Table 1). There were also interaction effects that significantly influenced nymph eclosion rates (Table 1). In a comparison of the interaction effects on nymph eclosion rates of treatment by plot location by year and by week, there were no significant differences in 2016 (Figure 6A). Nymph eclosion rates were only significantly different in the combination (control field by edge plot by July week 4 by 2017) when a single egg mass was collected and there was 100% parasitism from that egg mass with 0% nymph eclosion (Figure 6B). 

Egg parasitism rates increased over the season in both 2016 and 2017. In 2016, egg parasitism rates peaked in the first week of August with a mean ±SE of 24.7 ± 3.2% (Figure 7A). In 2017, egg parasitism rates peaked in the last week of August at 45.0 ± 7.2% (Figure 7B). In 2016, weekly nymph eclosion rates were the lowest in the first week of August when egg parasitism rates reached their peak (Figure 7A). In 2017, the mean (±SE) nymph eclosion rate was 49.5 ± 7.0% in the final week of August when egg parasitism rates reached 45.0% (Figure 7B).

### 3.3. Adult Parasitism

*Trichopoda pennipes* was the only adult parasitoid of squash bugs. There was no significant difference in the adult parasitism rate between fields with a flowering border and control fields (F = 1.1; df = 1, 7; *p* = 0.33). However, there was a significant difference between plot locations (F = 21.7; df = 1, 7; *p* = 0.002). There was a significant year by plot location effect (F = 9.8; df = 1, 7; *p* = 0.02). When the effects for plot location were evaluated separately for year by species effects, adult parasitism was only significantly higher in edge plots than in center plots for *A. tristis* adults in 2016 (F = 17.1; df = 1, 7; *p* = 0.004). Overall, adult parasitism was 24.6 ± 3.8%.

## 4. Discussion

The use of a flowering border did not influence egg parasitism rates. These results are consistent with a study that found no difference in the number of adult *G. pennsylvanicum* in squash fields with floral resources compared to squash fields without floral resources [17]. Adult *G. pennsylvanicum* feed on squash leaf trichome exudates, which provide an energy source for foraging adults [52]. Since parasitoids have a source of nutrition from squash leaves, it may not be necessary for them to forage for nectar from flowering plants.

The egg parasitoid, *G. pennsylvanicum*, was the predominant parasitoid, accounting for 93.9% of eggs parasitized. In previous studies, *G. pennsylvanicum* accounted for >99% of egg parasitism but egg parasitism rates by *G. pennsylvanicum* were substantially higher in those studies [41,44]. *Ooencyrtus anasae* was identified as the second most common parasitoid recovered from squash bug eggs. 

The identification of *Ooencyrtus* species is complicated by the lack of revisionary studies encompassing Nearctic taxa. Three species are known to attack *A. tristis* in the Nearctic: *O. anasae*, *O. californicus* Girault, and *O. papilionis* Ashmead. The latter two are quite different from *O. anasae.* Furthermore, *O. johnsoni* (Howard), another widespread species morphologically similar to *O. anasae*, was necessarily evaluated. However, it has not been recorded from *A. tristis* and has more metallic green reflections on the mesoscutum and metasoma, whereas *O. anasae* tends toward black coloration. A large integrative taxonomic study of the genus *Ooencyrtus* in Asia and North America is needed to conclusively differentiate between *O. anasae* and *O. johnsoni* [53].

In a previous study, egg parasitism rates by *G. pennsylvanicum* at BARC were much higher when squash was first planted in mid-May. In that case, the parasitism rate reached 29.6% by the second week of July, 50.5% by the third week, and 72.3% by the end of July [44]. In the current study, squash was planted in mid-late June and the parasitism rate was <1% in the second week in July and increased gradually, peaking in the end of August at 37.7%. After planting, squash bug egg masses increased rapidly. There were only eight squash bug egg masses collected during the first week of July compared with 83 egg masses in the second week and 263 egg masses in the third week. The results of the two studies at BARC suggest that *G. pennsylvanicum* populations primarily build up within the squash field through successive generations emerging from host egg masses and that it takes at least eight weeks after planting the squash for parasitoids to reach a population level high enough to parasitize 30% or more of squash bug eggs. 

Egg predation levels were very low (<1%). However, this study only measured eggs that were damaged by predators, not eggs that were removed by predators. In a previous study, plots in squash fields were monitored daily and freshly laid squash bug egg masses were marked and collected 48 h later. The number of intact/damaged eggs per mass was counted at the time the egg masses were marked and recounted at the time of collection. Within the 48-h period, 5.9% of marked egg masses had at least one egg removed by predators and 1.0% of marked egg masses were removed entirely by predators. Predators damaged/removed 5.2% of the total number of eggs collected from marked egg masses [44]. Therefore, egg predation was underestimated because egg/egg mass removal was not accounted for. Since 75% of the damaged eggs were found in squash fields with a flowering border, it is possible that the flowering border enhanced egg predation. However, further studies measuring both egg damage and egg removal by predators are necessary to evaluate the effect of a flowering border on egg predation.

The use of a flowering border did not result in lower squash bug nymph eclosion rates. In fact, nymph survival was slightly higher in squash fields with a flowering border. Overall, squash bug nymph survival was high. Average nymph eclosion rates only dropped <60% in the last week of August in 2017 when egg parasitism rates reached their peak.

Although *T. pennipes* was observed visiting buckwheat flowers (Cornelius, personal observation), adult parasitism rates did not differ between squash fields with and without a flowering border. Since *T. pennipes* is highly mobile, distances between fields with and without a border probably need to be greater to influence adult parasitism rates. However, adult *A. tristis* squash bugs collected in edge plots were significantly more likely to be parasitized than those collected in center plots in 2016, suggesting that *T. pennipes* is more likely to search for hosts on the edge of squash fields. Squash fields were located adjacent to corn, soybean, and pepper fields. *Trichopda pennipes* also parasitizes other leaf-footed and stink bug species [11,54]. Therefore, parasitic flies were most likely foraging widely for hosts within the agricultural landscape. Further studies of the host searching behavior of *T. pennipes* are needed to develop more effective methods of enhancing adult parasitism of squash bugs. 

*Anasa armigera* accounted for only 4.3% of egg masses, but 56.1% of the adults collected overall. Numbers of *A. armigera* egg masses collected from 2014–2018 have been consistently low, ranging from 2–4% of the total squash bug egg masses collected (Cornelius, unpublished). Also, nymphs of *A. armigera* are much less commonly seen in squash plots than *A. tristis* nymphs (Cornelius, personal observation). Although the reasons for the disparity in collections of adults and egg masses of *A. armigera* are unknown, it seems likely that differences in adult behavior could play a role. We collected any adult squash bugs that were found on squash plants when plants were searched for egg masses. We did not search for adult squash bugs hiding on the ground in the soil or under plant debris. If *A. armigera* adults are more likely to spend time on the plant compared with *A. tristis* adults, *A. armigera* adults were probably oversampled in relation to *A. tristis* adults. Most of the *A. armigera* adults were collected from 7–25 August 2017. In some incidences, multiple *A. armigera* adults were collected within one meter of each other, possibly due to aggregation behavior of the adults. Alternatively, adult *A. armigera* could be moving into squash fields late in the season. Currently, there are no field studies comparing the behavior of the two squash bug species. Further studies on the foraging and aggregation behavior of adult *A. armigera* squash bugs may shed light on the discrepancy between field collections of *A. armigera* adults and egg masses.

## 5. Conclusions

The proportion of people living in urban areas is projected to increase to 70% by 2050 [55]. Therefore, it is important to find ways to increase biodiversity and ecosystem services within urban habitats. As the rate of urbanization increases, urban agriculture will become increasingly more important for food security and for maintaining biodiversity [2,55]. The use of floral resources on urban farms is an important component of conserving pollinators and other beneficial insects. However, there is conflicting evidence concerning the effectiveness of floral resources for improving the biological control of pest populations. 

Flowering buckwheat did not increase parasitism rates of squash bugs in squash fields. The egg parasitoid, *G. pennsylvanicum,* was equally distributed in center and edge plots in fields with and without a flowering border. The egg parasitism rates increased gradually over the season, as successive generations of parasitoids emerged from host eggs within the squash fields. Overall, adult squash bugs collected in edge plots were more likely to be parasitized than those collected in center plots in fields with and without a flowering border. This difference was due to increased parasitism rates of *A. tristis* adults in edge plots in 2016.

Pollinators and other beneficial insects were frequently observed visiting the flowering buckwheat (Cornelius, personal observation). Therefore, the use of flowering buckwheat as a border around squash fields may have a positive effect on ecosystem services despite not augmenting parasitism rates against squash bugs. However, flowering plants could also attract herbivorous pests. Further research could examine the effects of flowering buckwheat on populations of other beneficial insects and herbivorous pests. It is also possible that other species of flowering plants or that planting strips of flowering plants between rows of squash could have a greater effect on biological control of squash bugs than using flowering plants as a border. Although floral resources play an important role in increasing populations of pollinators and other beneficial insects, their role in enhancing parasitoid populations in squash fields requires further investigation.

## Figures and Tables

**Figure 1 insects-10-00318-f001:**
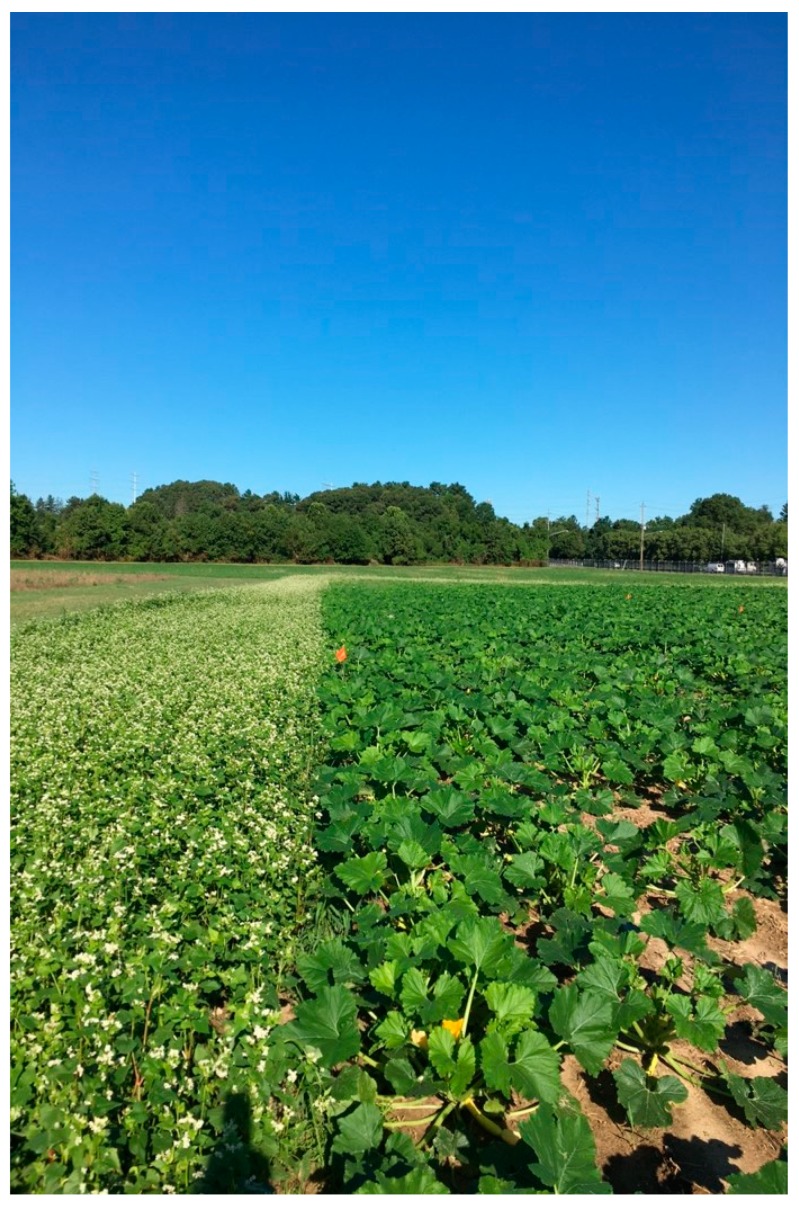
Squash field with a three-meter border of flowering buckwheat.

**Figure 2 insects-10-00318-f002:**
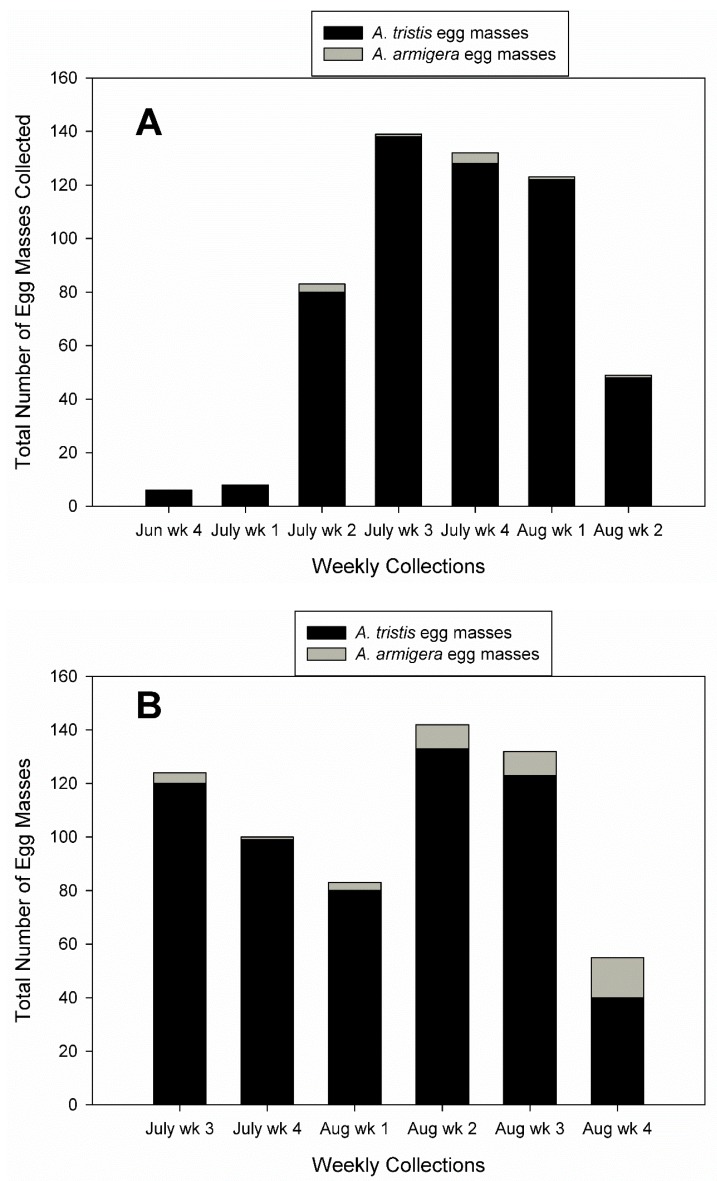
Total number of egg masses of *A. tristis* and *A. armigera* collected weekly from experimental plots in squash fields (**A**) 2016 and (**B**) 2017.

**Figure 3 insects-10-00318-f003:**
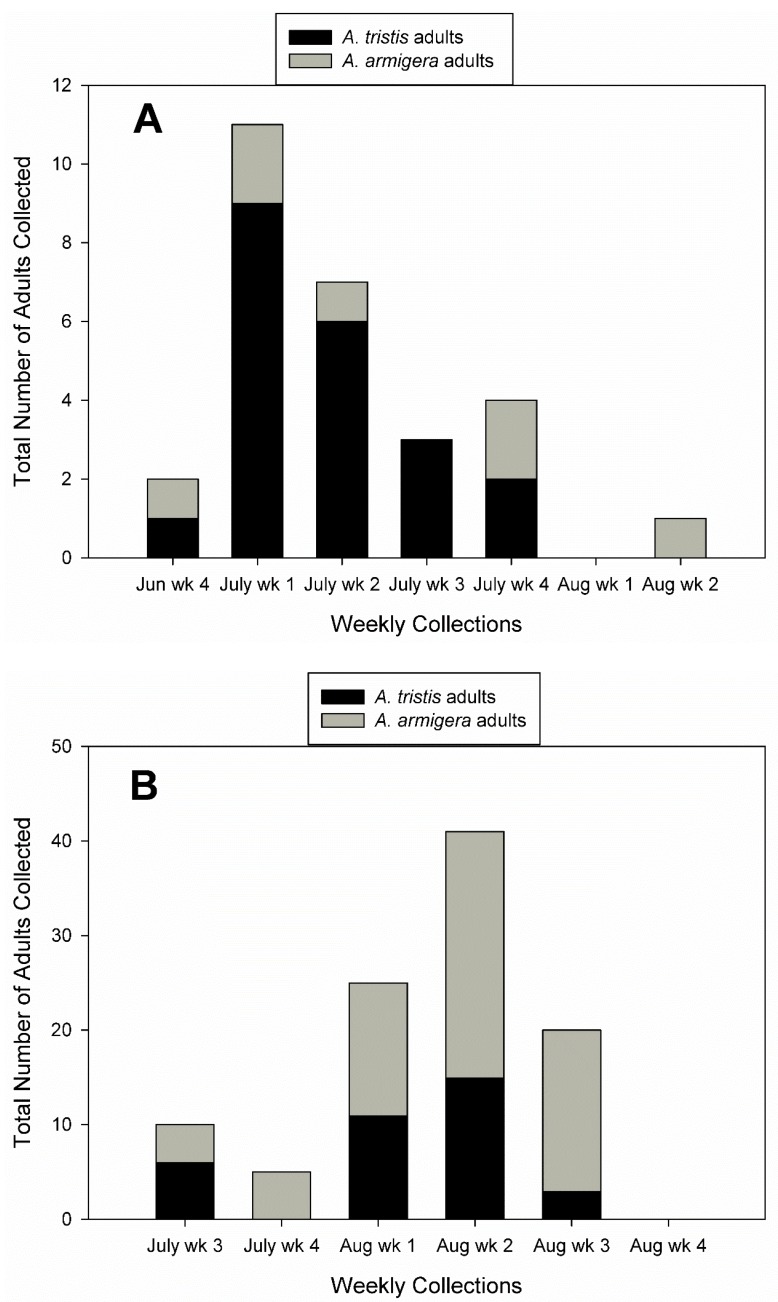
Total number of adults of *A. tristis* and *A. armigera* collected weekly from experimental plots in squash fields (**A**) 2016 and (**B**) 2017.

**Figure 4 insects-10-00318-f004:**
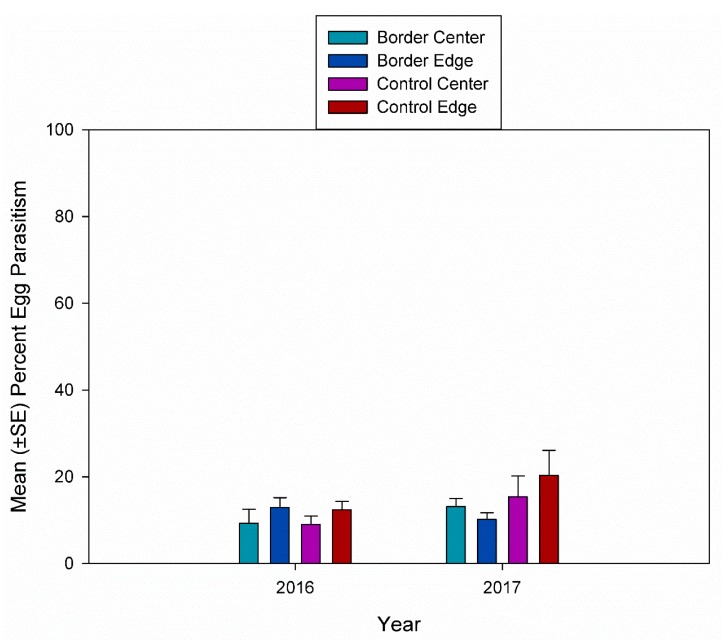
Mean (±SE) percent egg parasitism in center and edge plots in fields with a flowering border and without a flowering border from *A. tristis* egg masses collected in 2016 and 2017.

**Figure 5 insects-10-00318-f005:**
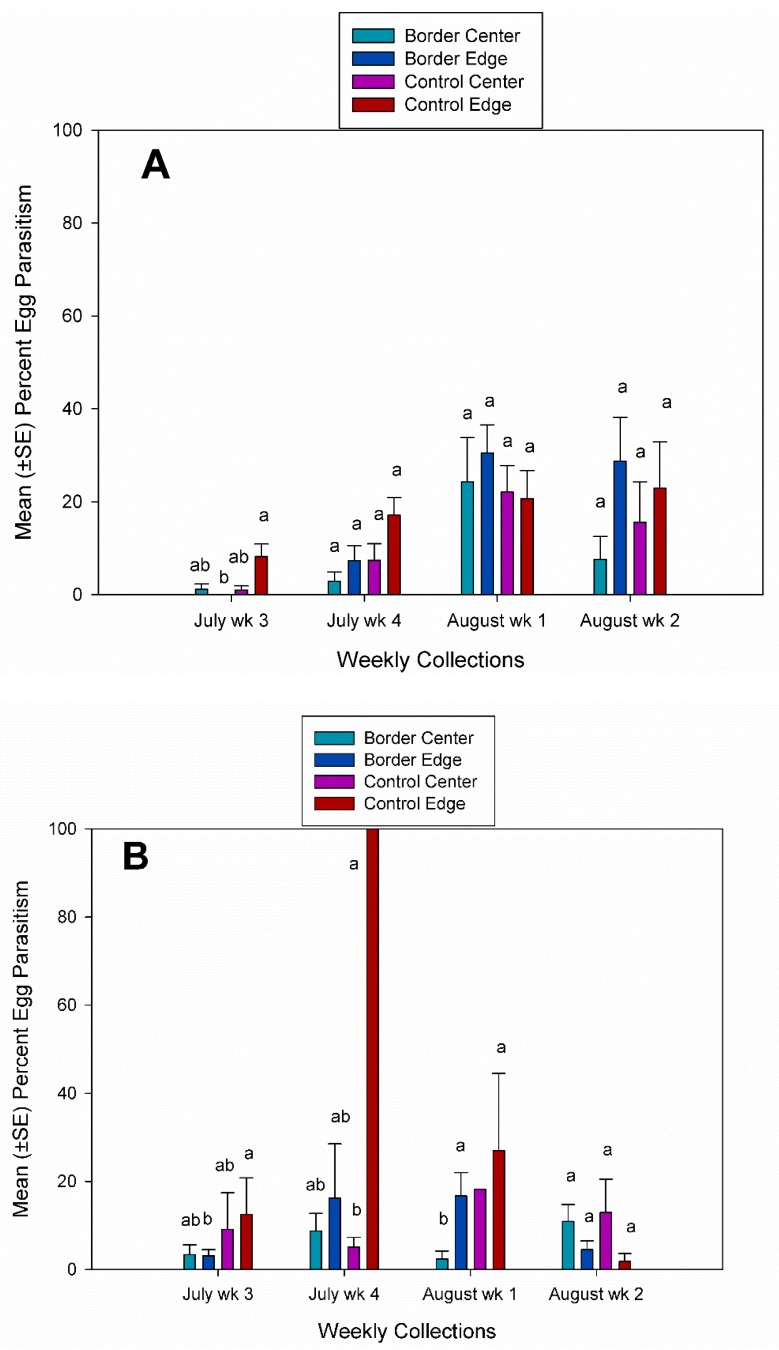
Mean (±SE) percent egg parasitism in center and edge plots in fields with a flowering border and without a flowering border from *A. tristis* egg masses collected each week. (**A**) 2016 and (**B**) 2017. Bars followed by a different letter within each week were significantly different (SAS PROC GLIMMEX: *p* < 0.05).

**Figure 6 insects-10-00318-f006:**
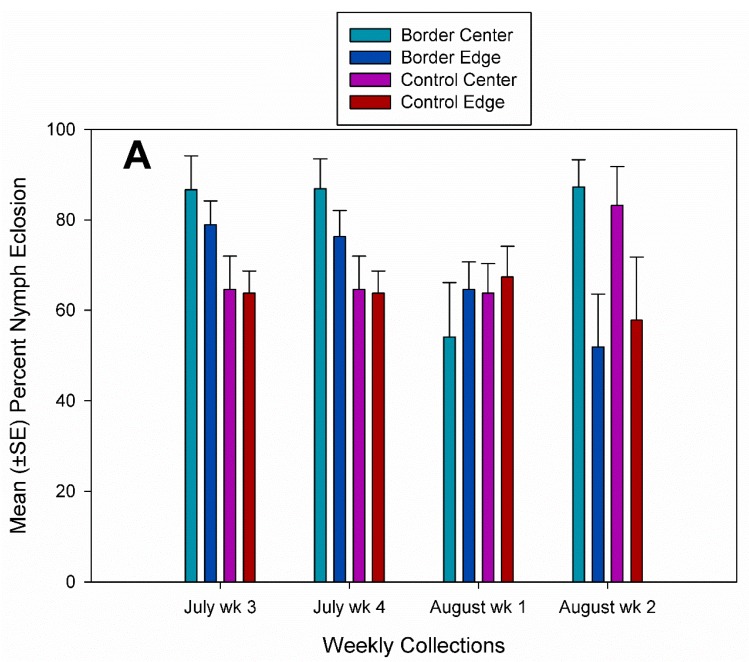
Mean (±SE) percent nymph eclosion in center and edge plots in fields with a flowering border and without a flowering border from *A. tristis* egg masses collected each week. (**A**) 2016 and (**B**) 2017. Bars followed by a different letter within each week were significantly different (SAS PROC GLIMMEX: *p* < 0.05).

**Figure 7 insects-10-00318-f007:**
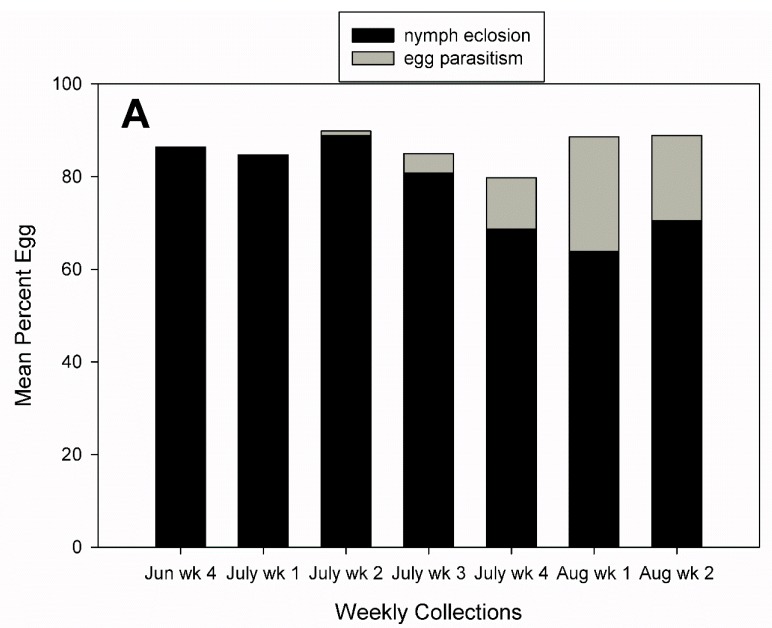
Weekly trends in mean percent nymph eclosion and egg parasitism of *A. tristis* eggs (**A**) 2016 (**B**) 2017.

**Table 1 insects-10-00318-t001:** F statistics and *p* values from a generalized linear mixed effects ANOVA using a negative binomial model with an offset equal to log number of eggs per mass. Effects of treatment (flowering border or control), plot location (center or edge), year (2016 or 2017), and week (July week 3–August week 2) on egg parasitism and nymph eclosion, were evaluated for *A. tristis* egg masses only.

Variable	Effect	F Statistic	df ^a^	*p* Value *
Egg Parasitism	Treatment	0.4	1	0.5
	Plot Location	3.7	1	0.09
	Year	0.6	1	0.47
	Week	21.5	3	<0.0001 *
	Treatment × Plot Location	1.6	1	0.24
	Year × Treatment	0.3	1	0.63
	Year × Plot Location	0.7	1	0.44
	Year × Week	11.4	3	<0.0001
	Treatment × Week	8.7	3	<0.0001 *
	Plot Location × Week	3.7	3	0.01 *
	Treatment × Year × Week	2.5	3	0.06
	Plot Location × Year × Week	15.1	3	<0.0001 *
	Treatment × Plot Location × Week	2.9	3	0.03 *
	Treatment × Plot Location × Year	5.1	1	0.05 *
	Treatment × Plot Location × Year × Week	12.8	2	<0.0001 *
Nymph Eclosion	Treatment	11.0	1	0.001 *
	Plot Location	12.0	1	0.0006 *
	Year	2.2	1	0.14
	Week	7.5	3	<0.0001 *
	Treatment × Plot Location	7.2	1	0.008 *
	Year × Treatment	9.3	1	0.002 *
	Year × Plot Location	5.4	1	0.02 *
	Year × Week	6.5	3	0.0002 *
	Treatment × Week	7.5	3	<0.0001 *
	Plot Location × Week	6.0	3	0.0005 *
	Treatment × Year × Week	5.2	3	0.002 *
	Plot Location × Year × Week	7.4	3	<0.0001 *
	Treatment × Plot Location × Week	5.9	3	0.0005 *
	Treatment × Plot Location × Year	7.5	1	0.006 *
	Treatment × Plot Location × Year × Week	6.7	3	0.0002 *

^a^ Degrees of Freedom for the numerator of the F statistic. The degrees of freedom for the denominator of egg parasitism is 8 for all model effects with a numerator of 1 degree of freedom and 638 for all model effects with a numerator of either 2 or 3 degrees of freedom. The degrees of freedom for the denominator of nymph eclosion is 836 for all model effects. * Significant interaction effects (*p* < 0.05).

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
