# Peer review of "Use of Flowering Plants to Enhance Parasitism and Predation Rates on Two Squash Bug Species Anasa tristis and Anasa armigera (Hemiptera: Coreidae)"

_insects, 2019, doi:10.3390/insects10100318_

Round 1
Reviewer 1 Report
Overall this was a nice and well written study. The authors investigated the effect of a flowering border of buckwheat on the parasitism and predation on two squash bug species within an urban farm setting. Predation rate of eggs was overall low, but parasitism was relatively high and increased as the season progressed and peaking in mid-August. The authors identified three egg parasitoids species, with the most common being Gryon pennsylvanicum. The flower treatment did not seem to impact parasitism of eggs or adults, but interestingly adult parasitism by tachinid flies was higher in plots along the edge.
I only have a couple of comments/edits.
Line 229: There are extra spaces before the period.
Line 596: It may be worth adding in a note either here or somewhere else in the conclusions mentioning the possibility of flowering plants to increase certain herbivores/pests as well.
Author Response
I removed the extra spaces. It was difficult to notice extra spaces, extra periods with the 'track changes" on. I added a sentence to the conclusion acknowledging that flowering plants could also attract herbivorous pests.
Reviewer 2 Report
The revisions have improved this paper. The main results are presented in Table 3. I think that it would be much better to present these results graphically. Additional specific comments are listed below.
Line numbers refer to the marked up manuscript <insects-590587-with track changes>.
Introduction
L 107 spelling of (Cucrbitales: Cucrbitaceae)
Results
L 251 Start the analysis be reporting whether week or year were significant factors. This determines whether you can make overall statements about nymph eclosion rate, egg parasitism rate, or the rate of egg mortality, or whether you need to analyze them within year, etc. Because week was significant, you should look at the other effects after accounting for week. I did not notice a statement regarding the significance of year.
L254 "Egg parasitism rates were similar in every plot" -- The data in Table 3 indicate highly variable rates of parasitism (range 0.2 - 94.87), so this conclusion is not well stated. Perhaps the measure of parasitism is too variable to measure any effect. Would weighting by the number of egg masses help?
L 269 - 276 This is the key analysis to the paper, but it appears to be missed. What are the differences in parasitism for these interactions (e.g., report the LS means for treatment after adjusting for week and year; likewise for plot location)?
L 275 "significant year by week interaction effects on egg parasitism rates in 2016 during July week 3 and in 2017 from July week 3-August week 1 (Table 4)." -- without looking at the means it is not clear what this means. Is this random variation, or was there a pattern? Reporting the means, or referring to a graph would be much clearer.
L277-280 Do you think that these differences are biologically meaningful?
L 580 - 590 Again, start with the overall model: were year and week significant? Report the LS means for treatment or plot location if analyzed within week within year.
Table 1 -- it is not clear to me why you present these results before those in Table 2. I do not see any discussion regarding the meaning of this information, which raises the question of why it is presented. Table 2 serves as the guide on how to test your hypotheses. You should look at treatment and plot location effects and their interaction within week within year.
The data in Table 3 would be better represented by two graphs (one for egg parasitism, the other for nymph eclosion), which would allow for visual comparison of the four treatment combinations over time.
Author Response
Table 3 has been converted into graphs. In the graphs, I used the actual means and standard errors. In Table 3, the means reported were estimated from the GLIMMIX model. I decided that it was better to show the actual means rather than the estimated means.
Table 1 was eliminated. Table 1 referred to differences between the two squash bug species. However, the number of A. armigera egg masses collected was very small and most of the A. armigera egg masses were collected in August of 2017. There were many weekly collections in specific plots with zero A. armigera egg masses. Therefore, I have eliminated the information about interaction effects between the two species. However, I have now reported the overall means in egg parasitism of the two squash bug species which show that the parasitoids attacked egg masses of the two species at a similar rate.
The hypothesis was that the flowering border would increase parasitism rates. For egg parasitism, the flowering border had no effect, not even a non-significant trend. Therefore, I have included a new graph of mean + standard error of egg parasitism on A. tristis egg masses for each treatment/plot location in each year (Fig. 4). Figure 4 illustrates visually that the flowering border had no effect. As reported in Table 1 (formerly Table 2), there were no significant differences between treatment, year, or plot location for egg parasitism rates on A. tristis egg masses.
I eliminated Table 4. The biologically meaningful effects were that egg parasitism increased over the season and that nymph eclosion was only reduced when egg parasitism reached its peak. However, the fields were not planted at the same time each year due to a rain delay in 2017. The statistical analysis was conducted from July week 3 - August week 2. Figure 7 was included to show the seasonal trends in nymph eclosion/ egg parasitism with the inclusion of all sampling dates in the two years.
Reviewer 3 Report
Very good scientifically sound study that provides some useful information on the influence of flowering plants on these important natural enemies of squash bug.
Author Response
Thank you for taking the time to review our manuscript.